# Seed Priming with *Devosia* sp. Cell-Free Supernatant (CFS) and Citrus Bioflavonoids Enhance Canola and Soybean Seed Germination

**DOI:** 10.3390/molecules27113410

**Published:** 2022-05-25

**Authors:** Ateeq Shah, Sowmyalakshmi Subramanian, Donald L. Smith

**Affiliations:** Department of Plant Sciences, McGill University, Montreal, QC H9X 3V9, Canada; ateeq.shah@mail.mcgill.ca (A.S.); sowmyalakshmi.subramanian@mcgill.ca (S.S.)

**Keywords:** flavonoids, cell-free supernatant, biostimulants, salt stress, seed priming, seed germination

## Abstract

Climate change, environmental pollution and associated abiotic stresses are beginning to meaningfully affect agricultural production worldwide. Salt stress is, however, one of the most important threats that significantly impairs plant growth and development. Plants in their early growth stages such as seed germination, seed emergence and early seedling growth are very sensitive to salt stress. Among the range of sustainable techniques adopted to improve seed germination and early plant growth is seed priming; however, with the use of ecofriendly substances, this is one of the most effective and economically viable techniques to improve seed tolerance against such environmental stresses. For instance, priming with appropriate non-synthetic compounds including microbial biostimulants are prominent ways to sustainably address these challenges. Therefore, in this research, by using the “priming technique”, two biostimulants were tested for their potential as sustainable approaches to improve canola and soybean seed germination under salt stress and optimal growth conditions. Canola and soybean seeds were primed with flavonoids extracted from citrus fruits (flavopriming) and cell-free supernatant (CFS; produced by a novel strain of *Devosia* sp.—SL43), alone and in combination, and exposed to low–higher levels of salt stress and ideal growth conditions. Both biostimulants showed promising effects by significantly improving seed germination of soybean and canola under both ideal and stressful conditions. However, increases in seed germination were greater under salinity stress as flavonoids and CFS with stress amelioration effects showed substantial and statistically significant improvements in seed germination under varying levels of salt stress. In addition, combinations (mixtures) of both biostimulants were tested to determine if their effects might be more additive or multiplicative than the individual applications. However, results suggested incompatibility of both biostimulants as none of the combinations showed better results than that of the individual applications of either flavonoids or CFS. Conceivably, the use of flavonoids and this novel *Devosia* sp. CFS could be significant plant growth enhancers, perhaps much better than the few other biostimulants and bacterial-based compounds currently in use.

## 1. Introduction

Seed germination is a key growth stage in a plant’s lifecycle; it plays a pivotal role in plant survival, crop yield and overall agricultural production [1,2]. There are several biotic and abiotic factors causing significant losses in seed germination and crop production. Soil salinity is an important abiotic stress as it reportedly causes substantial economic and production losses by adversely affecting field crop production [3]. The effects of salt stress begin from the very early stage of plants growth (seed germination and seedling growth) when plants are more susceptible to such environmental limitations [4]. Excess salts in the soil may diminish seed germination and seedling growth primarily through ion toxicity and water deficit (osmotic), thereby disrupting mobilization of stored reserves and affecting protein structure, and so causing immense damage to cells and tissues, eventually leading to reduced crop yield and production [5,6]. In addition, salinity (NaCl) stress delays seed germination by negatively regulating gibberellin synthesis and increasing ABA synthesis, leading to reduced productivity [7]. Oil seed crops such as canola and soybean are generally quite sensitive to stressful environmental conditions during seed germination as the oil in the seeds can oxidize during storage, which diminishes seed vigor [8]. Moreover, salinity stress may induce natural defense mechanisms in germinating seeds by initiating metabolic processes that may reduce ion toxicity or detoxify ROS (reactive oxygen species) through osmoregulation and antioxidation [9]. Sensitivity to salinity during early growth stages does not indicate sensitivity of mature plants. Many plant species are more resistant to salt stress during later stages of development than in their early growth stages, such as seed germination and seedling growth [10]. Therefore, studies should focus on stress resistance in a range of plant growth stages.

In order to develop adequate stress defenses, plants continuously modify their metabolism, rate of development, chemical defense systems and morphology [11]. However, plants’ inbuilt defense strategies are not always strong enough to resist or at least tolerate the effects of extreme environmental stresses to levels that could guarantee plant recovery and survival. This has led to the utilization of fertilizers, pesticides, and biostimulants and the use of advanced machinery in order to improve plant health, which eventually gave another direction to agricultural production. Although they have been effectively deployed against many biotic factors, enhanced challenges from abiotic stress factors associated with climate change are becoming a much bigger issue. That being said, we must now move toward “viable sustainability” by incorporating ecofriendly and appropriate organic techniques into our agricultural systems, while maintaining production capacity and global food security.

Flavonoids are polyphenolic compounds produced in plants as secondary metabolites; they play a pivotal role in plant growth and development even under conditions with extreme environmental limitations [12]. The role of flavonoids as growth stimulators, particularly as stress mediators, has been discussed extensively in many studies; however, their exogenous application has rarely been reported [13]. Likewise, phytomicrobiome and microbe-based products are now regarded as some of the best sustainable approaches for agricultural production, which are being referred as the drivers of “new green revolution” amid climate change scenarios [14]. For instance, thuricin 17 (TH17) and lipo-chitooligosaccharides (LCOs), two well-known bacteriocins, have been demonstrated to be substantially effective in plant growth stimulation and stress remediation, particularly under conditions of salt and drought stress [15,16,17].The application of microbes and their bioactive compounds have been overlooked at times but knowledge in this area is increasing, with them being one of the sustainable approaches in improving plant health under stressful conditions.

Therefore, in this study, a commercially used flavonoid extracted from citrus fruits and a cell-free supernatant from a novel bacteria *Devosia* sp. SL-43 (CFS) were evaluated for enhancement in seed germination of soybean and canola under ideal growth conditions and varying levels of salt stress. Among different seed application techniques, we opted for “seed priming” as this application method is a well-established technique in dry-land farming in tropical regions where the hot and dry climate not only cause water scarcity, but also make salinization prominent and its effects more intense. Priming seeds with flavonoids (flavopriming) and CFS will not only promote early seed germination but will also help seeds and/or seedlings to resist salt stress. The term “flavopriming” refers to priming seeds with flavonoids, and this is the first ever evaluation of this technology.

## 2. Results

### 2.1. Effect of Flavonoids on Seed Germination of Canola and Soybean

#### 2.1.1. Canola Seed Germination

Canola seed germination was significantly improved by flavopriming under both salt stress and optimal growth conditions; however, prominent and greater increases were detected under salt stress. Under optimal growth conditions, flavonoids significantly increased canola seed germination (after 18 h of sowing) by 16 and 9.7% as compared to unprimed (UP) and primed (P) control, respectively (*p* = 0.0001). At 24 h after the onset of seed germination, flavoprimed seeds had a significantly higher seed germination of 6 and 5.2% when compared against unprimed and primed controls. In the same way, significant increases were detected at 30 h and 36 h, with an increased seed germination of 5% when compared with both controls. Usually, the final germination remained similar among the treatments, but interestingly, one of the flavonoid treatments (1.5 µL g^−1^ seed) resulted in 5 % more final germination of seeds than the control treatments when seeds were either unprimed or primed with water (Figure 1A).

Interestingly, the seed germination increases were much greater under salt stress than optimal growth conditions showing salt stress amelioration effects in seeds treated with flavonoids. At 18 h, flavopriming substantially increased canola seed germination at 125 mM NaCl (*p* < 0.0001) by 82 and 256.8% over the primed and unprimed controls, respectively. However, in later recordings, flavoprimed treatments at 125 mM NaCl were ineffective in significantly increasing seed germination over the primed control; however, significant improvements over the unprimed control did occur (Figure 1B). Canola seed germination was substantially reduced with the increasing levels of salt stress; however, seeds treated with flavonoids were found to be less affected. For instance, at 150 mM NaCl, untreated seeds (at 18 h after sowing) had germination levels of 8.75 and 38.13% for unprimed and primed controls, respectively, while flavoprimed seeds had significantly greater seed germination at 10 times higher than that of the unprimed control (*p* < 0.0001) and two times greater than the primed control, indicating significantly improved salt tolerance. In the same way, flavonoid treatments at 24 h after sowing substantially improved canola seed germination, with the greatest increase being 37% over the unprimed control, while no significant differences were detected over the primed control. Similarly, although less pronounced, results were recorded at the later observation times (30 and 36 h after the onset of germination); however, significant increases were recorded for flavoprimed seeds when compared with unprimed control (Figure 1C).

Likewise, seed germination at 175 mM (NaCl), when recorded at 18 h, was substantially improved by flavonoid treatment (4.5 µL g^−1^ seed) with a seed germination of 67.5% (*p* < 0.0001), which was 26 times greater than the unprimed control and 163.3% higher than that of the primed control. However, flavonoid treatments had little effect at the final seed germination data collection time point (36 h) for canola, with no significant differences over both primed and unprimed controls (Figure 1D). The 200 mM NaCl treatment was extremely stressful for canola seeds as the germination was strongly inhibited in both unprimed (0.6%) and primed (6.3%) control treatments at 18 h after sowing; however, flavonoids, by elevating salt stress resilience, significantly increased the seed germination to 54.4%. At 24 and 30 h after sowing, canola seed germination was significantly improved by fourfold and 75.3%, respectively, when compared with the unprimed control. Likewise, at 36 h, the seed germination was significantly increased by 25% when compared against unprimed control (Figure 1E). (Data representing this text can be accessed from Appendix A at the end of the manuscript).

#### 2.1.2. Soybean Seed Germination

Flavonoids were less effective at increasing soybean seed germination under optimal growth conditions; there were no significant differences between flavonoid treatments and the primed control, indicating an effect similar to that of water, although with the flavonoid treatments, germination was 16% greater than the unprimed control at 18 h. At 24 and 48 h after sowing, no significant improvements could be detected for flavonoid treatments over both primed and unprimed controls (Figure 2A).

Flavonoids significantly increased soybean seed germination when exposed to varying levels of salt stress. Under 80 mM salt stress, soybean seed germination (at 24 h after sowing) was significantly greater than the unprimed control, by 392.2%, and, compared with the primed control, the germination was significantly higher, by 106.5% (Figure 2B). Flavonoid treatments were found to be more effective with increasing levels of salinity stress. For instance, at 100 mM NaCl, soybean seed germination at 24 h was significantly higher, by 383.3%, than the unprimed control, which occurred for flavonoid treatment (Fl 4.0 µL g^−1^ seed); compared with the primed control, soybean germination was significantly higher, by 163.6% (*p* < 0.0001) (Figure 2C). At 120 mM NaCl, the unprimed control failed to germinate at 24 h after sowing, which is reasonable given that soybean is very salt-sensitive. However, the primed control enhanced germination by 11.3% over the unprimed control. Flavonoid treatment (2.0 µL g^−1^ seed) with a seed germination of 58.7% was 58 times higher than the unprimed control and 4 times greater than that of the primed control (*p* < 0.0001) (Figure 2D). (Data accessibility for this section has been provided as supplements. Readers can refer to Appendix A).

### 2.2. Effects of Cell-Free Supernatant on Seed Germination of Canola and Soybean

#### 2.2.1. Canola Seed Germination

Under optimal conditions CFS did not strongly affect canola seed germination; no CFS dilution increased canola seed germination over the primed control, while all CFS treatments significantly elevated seed gemination when compared with the unprimed control. In contrast, CFS under salt stress (150 mM NaCl) significantly increased canola seed germination by 108% as compared to the primed control, and by 5 times for the unprimed control (*p* < 0.0001) at first recording (Figure 3). (Appendix A).

#### 2.2.2. Soybean SEED Germination

The CFS treatments did not increase soybean seed germination over the primed control; however, the highest dilution (1:1000) significantly increased seed germination over the unprimed control by 19% (*p* = 0.0004). Substantial increases were caused by CFS treatments at 48 and 72 h (11.1 and 8.1%, respectively) over the unprimed control, while none of the treatments caused significant increases over the primed control. In contrast, CFS application significantly increased soybean seed germination (at 24 h) under 100 mM NaCl stress by 54.5 and 183.3%, respectively, over primed and unprimed controls. (Appendix A).

### 2.3. Effects of Flavonoids and CFS Mixtures on Seed Germination of Canola and Soybean

#### 2.3.1. Canola Seed Germination

Among the five mixed biostimulant treatments, treatment with the highest dilution (4.5 µL g^−1^ seed + CFS 1:1000) significantly increased seed germination of canola under optimal growth conditions by 7 and 13% over primed and unprimed controls (*p* = 0.0035), respectively. However, the other four levels caused no increases after 18 h from onset of germination conditions. Substantial increases were detected by flavonoid plus CFS mixed treatments under salt stress. The most prominent was mix-3 (4.5 µL g^−1^ seed + CFS 1:250), which had the highest germination (86.9%) and was 128% greater than that of primed control and 563% over unprimed control at 18 h of seed germination (Figure 4). (Refer to Appendix A).

#### 2.3.2. Soybean Seed Germination

Soybean seed germination was, however, unaffected by the combined application of both biostimulants under optimal growth conditions; however, at 24 h, with the increase in CFS dilution, a progressive increase in soybean seed germination was detected, showing a strong correlation between CFS dilutions and soybean seed germination. Similar results occurred for soybean seed germination following the application of the same CFS dilution when applied alone. This leads to an assumption that a more diluted concentration of CFS, with or without flavonoids, may provide better results. Similarly, seed germination under salinity stress (100 mM NaCl) was highly affected by flavonoid plus CFS mixtures. The maximum germination after 24 h of sowing was recorded for 1.0 µL g^−1^ seed (flavonoid) plus 1:1000 (CFS) with a germination percentage of 56.3%, which was significantly greater than primed and unprimed controls by 104 and 275%, respectively. (Appendix A).

## 3. Discussion

### 3.1. Flavopriming

Flavonoids have been reported to be an important component of plant seeds, as they play significant roles in seed germination [18], including seed growth and development [19] and seed pigmentation [20]. The accumulation of flavonoids in seeds and seedlings during germination may be a strategic and programmed function carried out by the plants; however, in most cases, there is no clear understanding of the reason for these productions. In the case of legumes, it is well understood that a portion of this huge class of phenolics act as signaling compounds in phytomicrobial associations, which begin to accumulate in the seedlings within a week of seed germination [21]. However, along with these signals, other flavonoid classes are also exuded from seeds or roots; these serve many known and probably unknown functions. Certain classes of flavonoids exuded from germinating seeds and roots were found to have phyto-inhibitory effects, inhibiting seedling growth in a dose-dependent manner, where some may be stimulatory (at lower concentrations) or inhibitory (at higher concentrations) depending on the flavonoid type and class [22]. Leucocyanidin, a member of the leucoanthocyanidins (a class of flavonoids), was reported to have no impact on seed germination of *Brassica campestris* and *Lens esculenta*, whereas significant improvement was observed in seedling growth with the greatest promotion of root growth [23]. Similarly, flavonoids extracted from fruits and leaves of *V. rotundifolia* showed no effect on lettuce seedling hypocotyls, but increases of 138 and 169% in root growth of lettuce seedlings were observed following treatment with one of two flavonoid compounds [24].

Flavonoids, being strong antioxidants and a defensive chemical strategy of plants against unfavorable conditions including salinity, drought and UV radiation, are considered plant stress moderators. A study on Arabidopsis seed germination under salt stress suggested that the increased accumulation of flavonoids, as regulated by MYB111 (MYB transcription factor), reduced the adverse effects of salinity suggestively by altering salt tolerance through reactive oxygen species (ROS) scavenging [25]. There is very little information available in the literature regarding exogenous application of flavonoids on seed germination, and there is a huge gap in understating for these phenolics despite of their importance in plant growth and development. Hence, this research is an attempt to expand the understanding of this area by employing flavonoids using a “priming technique” (flavopriming) to determine its efficacy in seed germination enhancement of two major oil seed crops (canola and soybean) under optimal and salt stress conditions. It was observed that flavonoids with salt-mediating effects substantially improved seed germination of both crops at varying levels of salinity stress. Additionally, significant improvements were detected under optimal growth conditions, and these increases and variations between the treatments occurred very early in the seed germination process: 18 h for canola and 24 h for soybean. However, for data collected later, with significant differences against unprimed controls, most of the treatments showed insignificant differences over the primed control.

Usually, in such germination assays, almost all the seeds germinate by the time of the final germination measurement, irrespective of the treatments, unless the treatments inhibit germination. However, maximum seed germination was often attained much earlier for treated seeds than untreated seeds, depending on the efficacy of the biostimulants or the technique used. In this study, the final seed germination in most cases was not statistically significant among treatments, but the priming technique significantly improved seed germination at early measurements and treated seeds attained the final seed germination level much before the untreated seeds.

Interestingly, the seed germination responses of both crops to varying levels of flavonoids were found to be related to the growth conditions they were germinating under; seeds germinated under optimal growth conditions were more responsive to lower concentrations of flavonoids, while, under salt stress, significant increases were recorded with increasing levels of flavonoid concentrations. These variations were, however, not completely consistent among the treatments. For instance, under optimal growth conditions, the 1.5 µL g^−1^ seed, one of the lowest flavonoid concentrations used, resulted in significantly higher (the highest) seed germination (canola) at 24 h with a germination of 99.4%, followed by the 2.0 µL g^−1^ seed (97.5%), 2.5 µL g^−1^ seed (97.5%), 3.0 µL g^−1^ seed (96.9%) and 3.5 µL g^−1^ seed (95%). Interestingly, soybean seeds germinated at their maximum in the same range of flavonoid treatments as the highest soybean seed germination at 24 h; this was for the lowest concentration flavonoid treatments, 1.0 and 2.0 µL g^−1^ seed, as compared with the control and higher flavonoid levels.

These findings suggested that, under optimal growth conditions, crop plant seed germination not only responded better to lower flavonoid concentrations, but also indicated that both crops responded better to the same range of flavonoid concentrations. Under increasing levels of salt stress, flavonoid treatments with higher concentrations performed better than those with lower concentrations. For instance, under 200 mM NaCl (highest stress level), canola seed germination at 24 h was highest for the 5.0 µL g^−1^ seed with germination at 54.4%, consistently followed by the 4.5 µL g^−1^ seed (42.5%), 4.0 µL g^−1^ seed (30%), 3.5 µL g^−1^ seed (27.5%), 2.5 µL g^−1^ seed (20.6%) and 2.0 µL g^−1^ seed (10.6%). In the same way, soybean seed germination was significantly influenced by the highest flavonoid concentrations under varying levels of salt stress, except for 120 mM NaCl where treatments with lower concentrations (1.0 µL g^−1^ seed and 2.0 µL g^−1^ seed) caused the highest seed germination.

### 3.2. Cell-Free Supernatant

Seed germination is a key step in crop development as subsequent plant growth, development and yield are directly related to uniform seed germination and good seedling growth and establishment. There are several priming techniques used to improve seed germination of various plant species particularly under stressful conditions, for example, hydropriming [26], halopriming [27] and osmopriming [28]. However, priming seeds with biological materials or microbe-based products is not commonly performed. For example, certain strains of PGPR, when seeds are dressed or primed, or inoculated in the soil, are found to substantially improve seed germination and seedling growth of various crop species including radish [29], sunflower [30], chickpea [31], maize [32] and rice [33]. In addition to microbial strains, microbe-based compounds, for instance signals [34,35,36,37] and other VOCs (volatile organic compounds) [38,39,40], have been found to have stimulatory seed germination effects. In this part of the research, CFS prepared from cultures of a novel bacterial strain *Devosia* sp. (SL43) was used for priming. Seeds of canola and soybean were primed with various dilutions of the extracted material and evaluated for germination under optimum and salt-stress conditions.

Interesting results were obtained from the germination assays as significant improvements in seed germination of both crops resulted from CFS treatment, particularly under salt stress. Canola seed germination under optimal growth conditions was significantly improved by all CFS dilutions over the unprimed control; however, there were no significant differences between CFS treatments and the primed control. All CFS treatments responded more or less the same as there were only small differences among the CFS dilution levels. However, under salt stress (150 mM NaCl), one of the dilutions (CFS 1:250) resulted in the highest seed germination at 24 h after sowing, while the other treatments, either more concentrated or diluted, had much lower seed germination levels. However, seed germination at later measurement times was higher for the most diluted CFS treatments; for example, at 36 h after sowing, the final seed germination of canola for CFS 1:500 was significantly greater than both primed and unprimed controls, but less-diluted CFS treatments, with lower seed germinations, were not significantly different from either control.

The effects of CFS on soybean seed germination were consistent with canola seed germination; higher CFS dilutions elevated seed germination to a greater degree than lower dilutions. Under optimal growth conditions, the final seed germination of soybean (at 72 h) was higher for the most diluted CFS treatment: CFS 1:1000 (100%), followed by CFS 1:500 (96.3%), CFS 1:250 (92.5%) and CFS 1:100 (86.3%). Similarly, under salt stress, the highest seed germination levels for soybean were for the more diluted CFS treatments than at lesser dilutions. It seems that both canola and soybean are more responsive to lower concentrations or more diluted forms of CFS. Additionally, it generally seemed that flavonoids were more effective than CFS at elevating seed germination of canola and soybean under optimal growth conditions and varying levels of salt stress.

### 3.3. Combined Application of Flavonoids and CFS

In order to evaluate the combined effect of both biostimulants, the most effective flavonoid concentrations were chosen and combined with 5 CFS dilutions, making five treatments containing both flavonoids and CFS. The combined treatments significantly affected canola seed germination under optimal growth conditions when compared with both primed and unprimed controls, unlike the individual application of CFS. The combined application resulted in a significant increase in seed germination of canola by 13 and 7%, respectively, over unprimed and primed controls, and was not better than the individual application of flavonoids, which increased canola germination by 16 and 9.7% over unprimed and primed controls. This provides an understanding that the individual application of flavonoids under optimal growth conditions was not only better when applied alone, but its effects are more likely reduced when combined with CFS. Likewise, when germinating at 150 mM NaCl, the combined application of both biostimulants significantly improved canola seed germination over both controls; however, the germination increase was 10% lower than that of individual application of flavonoids and 5% greater than the individual application of CFS. Undoubtedly, despite the significant improvements in canola seed germination under salt stress, the combined application has somewhat reduced the effects of flavonoids alone.

In contrast, soybean seed germination under optimal growth conditions was unaffected by the combined application of both biostimulants unlike the individual applications of flavonoids and CFS. However, at 24 h, progressive enhancement of soybean seed germination occurred with the increasing CFS dilutions. For instance, the highest germination level occurred for 1.0 µL g^−1^ seed and CFS 1:1000, consistently followed by treatments containing lower dilutions/higher concentrations. Perhaps a more diluted concentration of CFS, >1:1000, might have more positively affected soybean seed germination. Under 100 mM NaCl, the combined application of both biostimulants, particularly treatment with the highest CFS dilution, significantly increased soybean seed germination over both primed and unprimed controls; however, the increases resulting from individual applications of either biostimulant were much higher than that of combined treatments. So, our findings strongly suggest that the individual application of both biostimulants are better than combined application, as the mixed levels in most cases had either no, equal or lower germination enhancement than the individual applications under both salt stress and optimal growth conditions, at least under the simple controlled environment conditions utilized here.

## 4. Material and Methods

### 4.1. Flavonoids

#### 4.1.1. Preparation of Flavonoid Solution for Seed Treatment

A stock solution containing 20% m/v flavonoids (extracted from citrus fruits) was prepared by dissolving 10 g of citrus flavonoid extract (Brand: Axenic, ordered from https://www.amazon.ca/gp/product/B07M9X4QJQ/ref=ppx_yo_dt_b_asin_title_o03_s00?ie=UTF8&psc=1, accessed on 15 June 2019) into 50 mL distilled water for pre-sowing seed treatment of canola and soybean. A range of flavonoid concentrations were assessed as there was little information guiding selection of doses and application procedures for seed treatments. In order to develop an adequate dose for seed treatment, very minute concentrations (0.5 to 5 µL g^−1^ seed) were used as treatments. The “µL g^−1^ seed” refers to the amount taken from the 20% flavonoid solution and applied on seeds based on their weight. Using the 0.5 µL g^−1^ seed as an example, for each gram of seed, the given amount (0.5 µL) from the stock solution was diluted with distilled water making a 2 mL flavonoid solution. For each gram of seed to be primed, 2 mL of solution was used.

#### 4.1.2. Seed Preparation and Priming

In a preliminary experiment, canola (*Brassica napus* L. var. L233P) and soybean (*Glycine max* L. var P09A62X) seeds were soaked in water for specific time periods to determine the time needed for seeds to imbibe the required amount of water to initiate germination. Soybean seeds were observed to imbibe water much more quickly than canola seeds; they imbibed to near the maximum level of water uptake within 2 to 3 h. Soaking soybean seeds for more than 5 to 6 h was observed to cause seed damage and reduce seed vigor. This was especially so when the seeds were dried quickly after soaking, which caused the soybean seed coat to be extensively damaged. However, canola seeds, having a hard seed coat, did not suffer damage across a range of soaking times (12, 24 and 48 h), and showed very little or no external morphological variation during preliminary experiments, except for increased seed size after imbibition. Based on these preliminary observations, canola and soybean seeds were decided to be soaked in flavonoid solutions at a proportion of 1:2 (seed (g): flavonoid solution (mL)) for 24 and 3 h, respectively. Before beginning with seed priming, canola and soybean seeds were surface-sterilized using 6% sodium hypochlorite for 5 min and then rinsed with sterile distilled water 2–3 times. Once sterilized, canola and soybean seeds were soaked in flavonoid solutions (flavoprimed) at given time durations. After that, the excess water was safely drained and seeds were washed with distilled water and air-dried until they reached their initial weight.

#### 4.1.3. Experimental Design and Treatments

The germination tests were carried out in 9 cm Petri plates (20 seeds per plate) incubated in a growth chamber in the dark at 25 ± 1 °C. For canola, 10 flavonoid concentrations ranging from 0.5 to 5 µL g^−1^ seeds were used. Twenty flavoprimed seeds per treatment were placed in Petri plates with filter paper soaked in 4 mL distilled water or NaCl solution for optimal and stressed conditions, respectively. In order to assess the efficacy of flavonoids against salt stress and to establish an adequate stress level for seed germination of canola, seeds were allowed to germinate at 4 salinity stress levels: 125, 150, 175 and 200 mM NaCl.

For soybean, the seed treatment was carried out in the same way as canola with the exception of reduced number of flavonoid levels. Soybean seeds were treated with five flavonoid concentrations, ranging from 1 to 5 µL g^−1^ seed. The evaluation of 5 flavonoid levels on soybean was selected from the results of canola seed germination experiments. We observed very little variations in seed germination of canola between close flavonoid concentrations. Therefore, by increasing the concentration gap between treatments, 5 levels of flavonoids, instead of 10, were used for soybean. For each treatment, 10 flavoprimed seeds were placed in Petri plates with filter paper soaked in 5 mL distilled water and NaCl solution for optimal and stressed growth conditions, respectively. The salt stress levels used for soybean germination were 80, 100 and 120 mM NaCl (refer to Table 1 for all the treatments tested).

### 4.2. Cell-Free Supernatant

#### 4.2.1. Propagation of Bacteria and CFS Extraction

*Devosia* sp. strain (SL43) was isolated from *Amphicarpaea bracteate* plants, growing wild along the shore of Lac St. Louis, Sainte-Anne-de-Bellevue, Quebec, Canada [41]. The bacterium was cultured in King’s medium B (Proteose peptone no. 3 20 g, K_2_HPO_4_ 0.66 g, MgSO_4_ 0.09 g, glycerol 0.06 mL, distilled water 1 L). The initial broth inoculum was taken from plated material and grown in 250 mL flasks containing 50 mL of the King’s B medium. The bacterium was grown for 48 h at 25 ± 2 °C on an orbital shaker at 150 rpm. The optical density was determined and adjusted to 0.1 using an Ultrospec 4300 Pro UV/Visible Spectrophotometer at 600 nm. Cell-free supernatant (CFS) was prepared by centrifuging the bacterial culture at 5000 g for 10 min (Awel^TM^ MF 48-R, NuAire, MN, USA). The supernatant was carefully decanted and clarified by vacuum filtration using a 0.22 μm filter. The CFS thus obtained was diluted to 1:50, 1:100, 1:250, 1:500, 1:1000 to be used as treatments.

#### 4.2.2. Experimental Design and Treatments

The estimation of canola and soybean seed germination, treated with CFS, was carried out following the same methods used for flavonoids application. However, after determining the adequate salt stress levels in the previous experiments, only one salt stress level was used for each crop: 150 and 100 mM NaCl for canola and soybean, respectively. Seed germination was recorded using the methods indicated previously (refer to Table 2 for treatments).

### 4.3. Flavonoids and Cell-Free Supernatant Combined

After individual applications of flavonoids and cell-free supernatant and obtaining reasonably viable results, we suspected the combined application of both biostimulants may provide better results than that of individual applications by performing synergistically against the salt stress. However, their compatibility was unknown. Therefore, this experiment was designed where both biostimulants were combined as mixtures and applied on seeds of canola and soybean. Based on the previous experiments, one of the best concentrations of flavonoids, indicated in both canola and soybean germination experiments, was combined with 5 CFS dilutions to make 5 levels of flavonoids + CFS mixtures. Seeds of canola and soybean grown under both optimal and stressed (salt) conditions were primed with mixed levels of both biostimulants at 1:2 (seed wt. (g): mix solution (mL)) proportions. The flavonoid concentrations selected to make “flavonoid + CFS mixes” for canola and soybean were 4.5 and 1.0 µL g-1 seed, respectively. The experimental design and growth conditions were the same as those used in CFS experiments (refer to Table 3 for treatments).

### 4.4. Data Collection

For canola, seed germination data were recorded at 6 h intervals, beginning 18 h after sowing. Seed germination was recorded until the maximum germination for all treatments was achieved, generally at 36 h after onset of the experiment. For soybean, data were recorded at 24 h intervals, beginning 24 h after sowing for 3 days. After recording the maximum germination, data were used to calculate the final seed germination percentage. All the experiments were repeated twice with 4 replications within each experiment. The seed germination experiments were conducted in Petri plates under controlled environmental growth conditions, where most of the seeds eventually germinate irrespective of whether seeds are grown under ideal or stressful conditions. So, the early seed germination recordings are very critical as most of the variations among treatments could be seen during initial stages, and therefore are given more focus in this experiment. In addition, the treatments were compared with both primed (water-soaked) and unprimed (unsoaked) control because the priming itself has stimulatory effects on seed germination irrespective of the compounds included. Therefore, by comparing flavoprimed and CFS-primed treatments with a water-primed control, increases in germination due to added biostimulants can be isolated; however, the overall technology would be the “biostimulant plus the priming effect”.

### 4.5. Data Analysis

All experiments were structured following a completely randomized design. The experimental datasets were pooled for data analysis using SAS 9.4 (SAS Institute Inc., Cary, NC, USA). Within this, germination data were analyzed by repeated measures using Proc GLM and Tukey’s multiple means comparison to determine statistical differences between treatments, at the 95% confidence level.

## 5. Conclusions

Overall, seed germination experiments indicated that there is a meaningful potential for flavonoids and CFS as seed germination enhancers for both canola and soybean, particularly under high levels of salt stress. Additionally, the results indicate the suitability of using both biostimulants as seed treatments via priming techniques. Flavopriming, a novel approach to treating seeds with flavonoids, provided much better enhancement of germination than CFS priming, and each biostimulant performed better when applied alone as compared to their combined application. Based on these findings, there is a substantial potential for both biostimulants as sustainable growth stimulators. Since these products are under evaluation and have been used for the first time in a germination assay, they need further evaluation using the same or different seed treatment techniques, optimizing each for specific crops, and to determine their compatibility with other major crops. Since both products, by inducing stress resistance in germinating seeds, caused greater seed germination improvement under salinity stress, there is a need to determine efficacy against other major abiotic stressors, such as drought and low- and high-temperature conditions, because they may have the potential to induce resistance against other stresses, particularly drought, as drought and salt stress exert similar detrimental effects on crops, and plant responses to them are often quite similar.

## Figures and Tables

**Figure 1 molecules-27-03410-f001:**
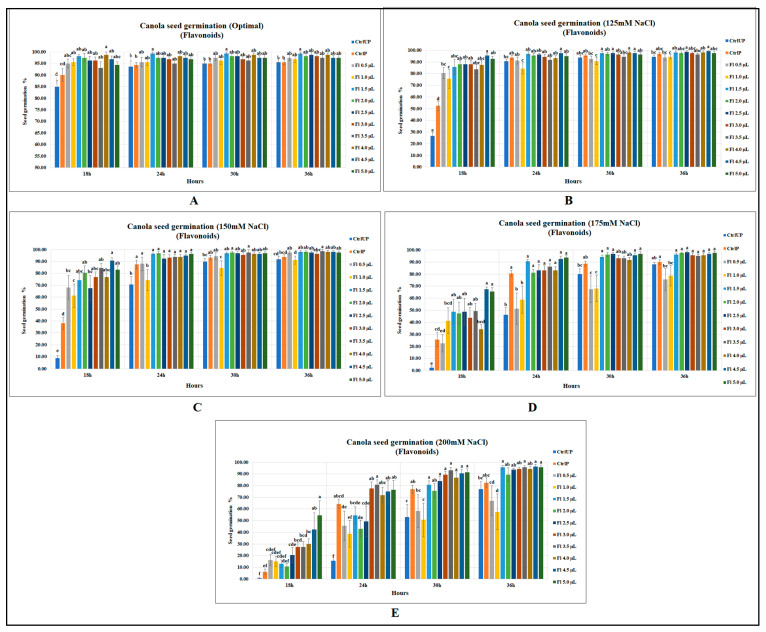
Flavopriming and seed germination of canola (**A**) optimal growth conditions (**B**) 125 mM NaCl (**C**) 150 mM NaCl (**D**) 175 mM NaCl (**E**) 200 mM NaCl. Data shown as mean ± SE of 8 biological replicates; different letters indicate values determined by Tukey’s multiple mean comparison to be significantly different (*p* < 0.05) among treatments.

**Figure 2 molecules-27-03410-f002:**
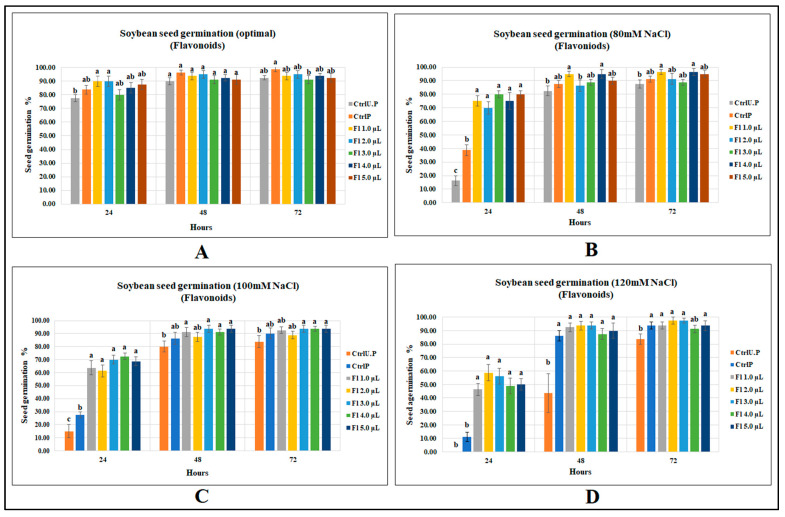
Flavopriming and seed germination of soybean: (**A**) optimal growth conditions, (**B**) 80 mM NaCl, (**C**) 100 mM NaCl, (**D**) 120 mM NaCl. Data shown as mean ± SE of 8 biological replicates; different letters indicate values determined by Tukey’s multiple mean comparison to be significantly different (*p* < 0.05) among treatments.

**Figure 3 molecules-27-03410-f003:**
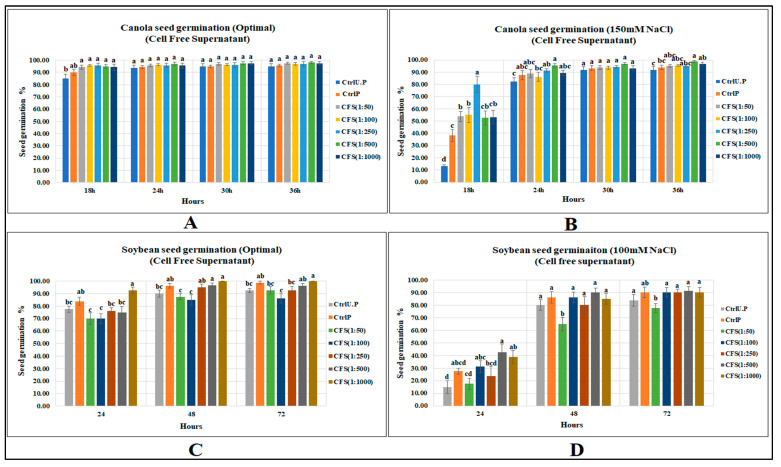
CFS affecting seed germination of canola and soybean: (**A**) Canola (optimal), (**B**) Canola (150 mM NaCl), (**C**) Soybean (optimal), (**D**) Soybean (100 mM NaCl). Data shown as mean ± SE of 8 biological replicates; different letters indicate values determined by Tukey’s multiple mean comparison to be significantly different (*p* < 0.05) among treatments.

**Figure 4 molecules-27-03410-f004:**
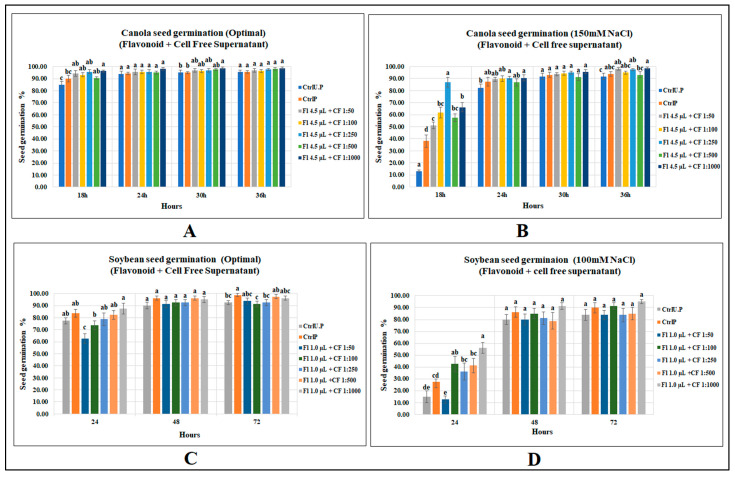
Effect of flavonoids and CFS mixtures: (**A**) canola (optimal growth), (**B**) canola (150 mM NaCl), (**C**) soybean (optimal), (**D**) soybean (120 mM NaCl). Data shown as mean ± SE of 8 biological replicates; different letters indicate values determined by Tukey’s multiple mean comparison to be significantly different (*p* < 0.05) among treatments.

**Table 1 molecules-27-03410-t001:** Flavonoid and salt stress levels in canola and soybean experimentation.

Canola		Soybean	
Flavonoid Levels	Salt Stress (NaCl) Levels	Flavonoid Levels	Salt Stress (NaCl) Levels
0.5 µL g^−1^ seed 1.0 µL g^−1^ seed 1.5 µL g^−1^ seed 2.0 µL g^−1^ seed 2.5 µL g^−1^ seed 3.0 µL g^−1^ seed 3.5 µL g^−1^ seed 4.0 µL g^−1^ seed 4.5 µL g^−1^ seed 5.0 µL g^−1^ seed	125 mM NaCl 150 mM NaCl 175 mM NaCl 200 mM NaCl	1.0 µL g^−1^ seed 2.0 µL g^−1^ seed 3.0 µL g^−1^ seed 4.0 µL g^−1^ seed 5.0 µL g^−1^ seed	80 mM NaCl 100 mM NaCl 120 mM NaCl

Flavonoid and salt stress levels were combined in a factorial manner. Two controls (primed-water soaked and unprimed), each for both optimal and stressed growth conditions, were compared with the flavonoid treatments.

**Table 2 molecules-27-03410-t002:** Cell-free supernatant treatments for canola and soybean seed germination.

Experimental Units (Seed Germination-Cell-Free Supernatant)			
Canola		Soybean	
Unstressed	Stressed (NaCl)	Unstressed	Stressed (NaCl)
Control—I	Control—I	Control—I	Control—I
(Unprimed)	150 mM NaCl (Unprimed)	(Unprimed)	100 mM NaCl (Unprimed)
Control—II	Control—II	Control—II	Control—II
(hydroprimed)	150 mM NaCl (hydroprimed)	(hydroprimed)	100mM NaCl (hydroprimed)
CFS 1:50	CFS 1:50 + 150 mM NaCl	CFS 1:50	CFS 1:50 + 100 mM NaCl
CFS 1:100	CFS 1:100 + 150 mM NaCl	CFS 1:100	CFS 1:100 + 100 mM NaCl
CFS 1:250	CFS 1:250 + 150 mM NaCl	CFS 1:250	CFS 1:250 + 100 mM NaCl
CFS 1:500	CFS 1:500 + 150 mM NaCl	CFS 1:500	CFS 1:500 + 100 mM NaCl
CFS 1:1000	CFS 1:1000 + 150 mM NaCl	CFS 1:1000	CFS 1:1000 + 100 mM NaCl

**Table 3 molecules-27-03410-t003:** Flavonoids and cell-free-supernatant-combined treatments for canola and soybean seed germination.

Experimental Units (Seed Germination-Flavonoids and Cell-Free Supernatant Combined)			
Canola		Soybean	
Unstressed	Stressed (NaCl)	Unstressed	Stressed (NaCl)
Control—I	Control—I	Control—I	Control—I
(Unprimed)	150 mM NaCl (Unprimed)	(Unprimed)	100 mM NaCl (Unprimed)
Control—II	Control—II	Control—II	Control—II
(hydroprimed)	150 mM NaCl (hydroprimed)	(hydroprimed)	100 mM NaCl (hydroprimed)
4.5 µL + CFS 1:50	4.5 µL + CFS 1:50 (150 mM NaCl)	1.0 µL + CFS 1:50	1.0 µL + CFS 1:50 + 100 mM NaCl
4.5 µL + CFS 1:100	4.5 µL + CFS 1:100 (150 mM NaCl)	1.0 µL + CFS 1:100	1.0 µL + CFS 1:100 + 100 mM NaCl
4.5 µL + CFS 1:250	4.5 µL + CFS 1:250 (150 mM NaCl)	1.0 µL + CFS 1:250	1.0 µL + CFS 1:250 + 100 mM NaCl
4.5 µL + CFS 1:500	4.5 µL + CFS 1:500 (150 mM NaCl)	1.0 µL + CFS 1:500	1.0 µL + CFS 1:500 + 100 mM NaCl
4.5 µL + CFS 1:1000	4.5 µL + CFS 1:1000 (150 mM NaCl)	1.0 µL + CFS 1:1000	1.0 µL + CFS 1:1000 + 100 mM NaCl

## Data Availability

The data presented in this study are openly available in FigShare at DOI:10.6084/m9.figshare.19809010.

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
