# Peer review of "Seed Priming with Devosia sp. Cell-Free Supernatant (CFS) and Citrus Bioflavonoids Enhance Canola and Soybean Seed Germination"

_molecules, 2022, doi:10.3390/molecules27113410_

Round 1
Reviewer 1 Report
In the manuscript entitled "Seed priming with Devosia sp. cell free supernatant (CFS) and citrus bioflavonoids enhance canola and soybean seed germination." the authors investigated the effect of flavonoid and CFS on seed germination of two important agricultural plants in ideal conditions and under salt stress. The research work is very well-planed and performed, the chosen methods and system is adequate for this research. The methods were described in detail, so the experiments can be repeated by others and the presentation of the results are clear, the figures and tables are well-prepared. In general, the manuscript is written in good manner and order. Only some disturbing Chinese characters appeared in the text instead of referring the figures in the section 3.2. Experimental Design and treatments, and in the section 4. Results.
Author Response
Reviewer 1
Reviewer 1
In the manuscript entitled "Seed priming with Devosia sp. cell free supernatant (CFS) and citrus bioflavonoids enhance canola and soybean seed germination." the authors investigated the effect of flavonoid and CFS on seed germination of two important agricultural plants in ideal conditions and under salt stress. The research work is very well-planed and performed, the chosen methods and system is adequate for this research. The methods were described in detail, so the experiments can be repeated by others and the presentation of the results are clear, the figures and tables are well-prepared. In general, the manuscript is written in good manner and order. Only some disturbing Chinese characters appeared in the text instead of referring the figures in the section 3.2. Experimental Design and treatments, and in the section 4. Results.
Authors Response
Thank you for your detailed insight. However, I couldn’t find any “Chinese characters” in the captions or figure cross-references.
Reviewer 2 Report
The authors present a study on the use of seed priming with flavonoids, novel bacterial derived compounds (cell free supernatant) and their combination to improve seed germination under optimal and stress (salt) growth conditions in two oil crop species. The manuscript is relevant for the field of interest of the Special issue since opens to new perspectives on use of flavonoids as sustainable priming agents to promote seed germination. However, the initial hypothesis is not fully supported by the presented results and hence further experiments are needed.
General comments
The authors stated that seed priming with two biological compounds affects seed germination during salt stress, however, the assessment of the germination characteristics alone seems not to be enough. I could suggest integrating the presented work by analysing the effects of the two potential priming agents on physiological performance (e.g. early seedling growth) at the different salinity levels. In addition, the activation of non–enzymatic and enzymatic antioxidant machinery as well as oxidative stress parameters such as malondialdehyde (MDA) level in (un)primed seeds could be evaluated.
Specific comments
Materials and methods as well as results require improvement in multiple areas for clarity and reproducibility.
Section Materials and Methods
- Please provide details on plant species and seed sterilization procedures
- In the experimental design a table for the applied flavonoids and CFS combinations could be added
Section Results
- A representative image of the (un)primed seeds before and after different salt stress treatments could be added
- Details about statistical analysis are missing in the figures
- The CFS effect on soybean seed germination was not reported in the figures
- The supplementary tables were not mentioned in the manuscript
Author Response
Reviewer 2
Reviewer 2
The authors present a study on the use of seed priming with flavonoids, novel bacterial derived compounds (cell free supernatant) and their combination to improve seed germination under optimal and stress (salt) growth conditions in two oil crop species. The manuscript is relevant for the field of interest of the Special issue since opens to new perspectives on use of flavonoids as sustainable priming agents to promote seed germination. However, the initial hypothesis is not fully supported by the presented results and hence further experiments are needed.
General comments from Reviewer 2
Comment 1:
The authors stated that seed priming with two biological compounds affects seed germination during salt stress, however, the assessment of the germination characteristics alone seems not to be enough. I could suggest integrating the presented work by analyzing the effects of the two potential priming agents on physiological performance (e.g. early seedling growth) at the different salinity levels. In addition, the activation of non–enzymatic and enzymatic antioxidant machinery as well as oxidative stress parameters such as malondialdehyde (MDA) level in (un)primed seeds could be evaluated.
Authors Response:
Thank you for your suggestions. Since this was the first time these bioproducts were under evaluation, we opted for a germination assay as an “initial” to identify their potential as biostimulants. After having promising effects, we proceeded with the second step which is plant growth responses (early plant growth) on application of these biostimulants under both ideal growth conditions and salt stress. However, work on this part is still in progress!
Specific comments from Reviewer 2
Comment 2:
Please provide details on plant species and seed sterilization procedures
Authors response:
As advised, plant species and sterilization procedures have been given in the text.
Comment 3:
In the experimental design a table for the applied flavonoids and CFS combinations could be added
Authors response:
A table containing treatments for combined application of flavonoids and CFS has been added in the indicated section.
Reviewer 2 comments from “Results”
Comment 4:
The CFS effect on soybean seed germination was not reported in the figures
Authors response:
A figure related to this has been added.
Comment 5:
The supplementary tables were not mentioned in the manuscript
Authors response:
Supplementary tables have been mentioned in the text, as advised.
Reviewer 3 Report
The reviewed manuscript is attached and some statements should be re-evaluated.

Author Response
Reviewer 3
Comment 1:
Source needed for the second part of the statement
Statement
“In order to develop adequate stress defenses, plants continuously modify their metabolism and rate of development, their chemical defense systems and/or morphology”.
Authors response:
Reference added in support of the statement
Bano, C., Amist, N. and Singh, N.B. (2019). Morphological and Anatomical Modifications of Plants for Environmental Stresses. In Molecular Plant Abiotic Stress (eds A. Roychoudhury and D. Tripathi). https://doi.org/10.1002/9781119463665.ch2
Comment 2:
Too many factors have been included here, so this could be avoided
Statement
However, in most cases such adaptations are not enough to allow plants to cope with the extreme biotic and abiotic challenges during growth. This led to the development of pesticides, which were included in “green revolution” during the 20th century
Authors response:
The text has been rephrased as:
“However, plants inbuilt defense strategies are not always strong enough to eliminate or at least reduce the effects of extreme environmental stresses to levels that could guarantee plant recovery and survival. This led to the development of fertilizers, pesticides, and use of advanced machinery of times which gave another direction to agricultural production
Comment 3:
This part could be shorter
Paragraph
In this study, two biological compounds, flavonoids and a bacterial based cell free solution, were investigated as sustainable biostimulants to promote early plant growth of canola and soybean under both optimal and salt stress conditions. The role of flavonoids as growth stimulators, particularly, as stress mediators has been discussed extensively in many studies, however, their exogenous application has rarely been reported [8]. Likewise, the role of the phytomicrobiome and microbe based products is now regarded as one of the best sustainable approaches for agriculture; they are being referred as the “new green revolution” [9]. There is potentially an array of microbe based compounds, mostly signals; for instance, thuricin 17 (TH17) and lipo-chitooligosaccharides (LCOs) that have been demonstrated to be substantially effective in plant growth stimulation and stress remediation; particularly under salt and drought stress conditions
Authors response:
The text has been shortened. Changes could be seen in “track changes”
In addition to the comments mentioned above, reviewer 3 has highlighted some typos, grammatical mistakes, and unclear statements, which have been corrected accordingly. In addition, we have proofread the manuscript to check for further changes and improvements. Please see the changes in “track changes”
Round 2
Reviewer 2 Report
Thank you for the revised version of your manuscript. I only have the following minor comments to address:
Section Materials and Methods
Lines 47-48: Check if the link is correct.
Section Results
-Figures: legend could be completed by adding details about statistical analysis.
- § 3.4 Supplementary data: please revised the text according to numbering of the supplementary tables. See the indication reported in the journal template for the correct position of this section in the manuscript.
Please carefully revised the manuscript there are still few spelling and typos errors, as example “Leucocanidin” is Leucocyanidin.
Author Response
Reviewers’ comments and response (Round 2)
Reviewer 2
Reviewer 2
Thank you for the revised version of your manuscript. I only have the following minor comments to address:
Comment 1
Lines 47-48: check if the link is correct
Authors response
Link to the flavonoid product has been checked and verified.
Comment 2
Figures: legend could be completed by adding details about statistical analysis
Authors response
Figure legends have been updated as suggested.
Comment 3
3.4 Supplementary data: please revised the text according to numbering of the supplementary tables. See the indication reported in the journal template for the correct position of this section in the manuscript.
Authors response
The section has been positioned at the end of the manuscript as per MDPI template. In addition, considering the complexity of revising the whole text according to the numbering of the tables provided, we have revised the numbering of tables according to the text presented. Supplementary tables have been reordered and renumbered appropriately. Hope that will suffice to address the reviewer’s concern.
Comment 4
Please carefully revised the manuscript there are still few spelling and typos errors, as example “Leucocanidin” is Leucocyanidin.
Authors response
Thank you for highlighting this! We have given another read to the manuscript. Hope there are no more typos left.